

# Identifying suitable reference genes for gene expression analysis in developing skeletal muscle in pigs

Guanglin Niu[1,2,*], Yalan Yang[1,2,*], YuanYuan Zhang[1], Chaoju Hua[1], Zishuai Wang[1], Zhonglin Tang[1,2] and Kui Li[1,2]

[1] The Key Laboratory for Domestic Animal Genetic Resources and Breeding of Ministry of Agriculture of China, Institute of Animal Science, Chinese Academy of Agricultural Sciences, Beijing, China
[2] Agricultural Genome Institute at Shenzhen, Chinese Academy of Agricultural Sciences, Shenzhen, China
[*] These authors contributed equally to this work.

Corresponding author
Zhonglin Tang,
zhonglinqy_99@sina.com,
tangzhonglin@caas.cn

## ABSTRACT

The selection of suitable reference genes is crucial to accurately evaluate and normalize the relative expression level of target genes for gene function analysis. However, commonly used reference genes have variable expression levels in developing skeletal muscle. There are few reports that systematically evaluate the expression stability of reference genes across prenatal and postnatal developing skeletal muscle in mammals. Here, we used quantitative PCR to examine the expression levels of 15 candidate reference genes (*ACTB*, *GAPDH*, *RNF7*, *RHOA*, *RPS18*, *RPL32*, *PPIA*, *H3F3*, *API5*, *B2M*, *AP1S1*, *DRAP1*, *TBP*, *WSB*, and *VAPB*) in porcine skeletal muscle at 26 different developmental stages (15 prenatal and 11 postnatal periods). We evaluated gene expression stability using the computer algorithms geNorm, NormFinder, and BestKeeper. Our results indicated that *GAPDH* and *ACTB* had the greatest variability among the candidate genes across prenatal and postnatal stages of skeletal muscle development. *RPS18*, *API5*, and *VAPB* had stable expression levels in prenatal stages, whereas *API5*, *RPS18*, *RPL32*, and *H3F3* had stable expression levels in postnatal stages. *API5* and *H3F3* expression levels had the greatest stability in all tested prenatal and postnatal stages, and were the most appropriate reference genes for gene expression normalization in developing skeletal muscle. Our data provide valuable information for gene expression analysis during different stages of skeletal muscle development in mammals. This information can provide a valuable guide for the analysis of human diseases.

## INTRODUCTION

Gene expression analysis provides important information for the study of gene function. Reference genes are used to judge gene expression levels and changes in target gene expression. Quantitative PCR (qPCR) is an important method in evaluating gene expression, which was first invented by Applied Biosystems Corporation (USA) in 1996. This technology significantly advanced gene quantitative research, and provided high sensitivity, specificity, and accuracy (*Mackay, 2004*; *Valasek & Repa, 2005*). Quantitative

PCR analysis can be used to explore differences in gene expression at different developmental periods or under different conditions. The selection of appropriate reference genes for qPCR analysis can improve the accuracy and reproducibility of the study by normalizing the expression of target genes with respect to the expression of a selected standard gene (*Huggett et al., 2005*). However, the qPCR method can be affected by reaction parameters such as template quality, operating errors, and amplification efficiency (*Bustin, 2002*; *Gabert et al., 2003*; *Ginzinger, 2002*; *Vandesompele et al., 2002*; *Wolffs et al., 2004*; *Yeung et al., 2004*) Thus, qPCR data should be normalized with respect to one or more constitutively expressed reference or housekeeping genes, which corrects for experimental variability in some parameters.

Reference genes have to be validated and consistently expressed under various circumstances. Widely used reference genes for expression analyses include *GAPDH*, *ACTB*, and *HPRT* (*Blaha et al., 2015*; *Boosani, Dhar & Agrawal, 2015*; *Wang et al., 2016*; *Zhang et al., 2016b*; *Zhao et al., 2015*). These genes are reported to have consistent expression levels under various conditions such as different organs and different developmental stages (*Tang et al., 2007*). However, expression of these selected reference genes has not proven to be as stable as originally presumed, and their expression can be highly variable under different conditions (*Jain et al., 2006*; *Wan et al., 2010*; *Wang et al., 2015*). Therefore, qPCR has been used to identify appropriate reference genes in humans (*Andersen, Jensen & Orntoft, 2004*; *Warrington et al., 2000*), animals (*McCulloch et al., 2012*; *Martinez-Giner et al., 2013*; *Robledo et al., 2014*; *Tatsumi et al., 2008*), and plants (*Hu et al., 2009*; *Huis, Hawkins & Neutelings, 2010*; *Jain et al., 2006*; *Zemp, Minder & Widmer, 2014*). In addition, the use of more than one reference gene might be necessary to accurately normalize gene expression levels and avoid relative errors (*Jian et al., 2008*; *Ohl et al., 2005*).

Skeletal muscle development is an important subject of biological research, and it plays an important role in meat production and various diseases (*Li et al., 2016a*; *Nixon et al., 2016*; *Obata et al., 2016*; *Zabielski et al., 2016*; *Costa Junior et al., 2016*; *Fonvig et al., 2015*; *Putti et al., 2015*; *Thivel et al., 2016*). Studies of muscle development often explore gene expression under different conditions (*Krist et al., 2015*; *Wang, Xiao & Wang, 2016*; *Zhang et al., 2016a*; *Zhang et al., 2015*). The expressions of most genes display variable expression levels in prenatal and postnatal periods, and reference genes that are frequently used for other experiments are not suitable for studies of skeletal muscle development. Several studies have been conducted to select reference genes in pigs (*Li et al., 2016b*; *Monaco et al., 2010*; *Park et al., 2015*; *Zhang et al., 2012*; *McCulloch et al., 2012*). However, few studies have focused on reference genes in developing skeletal muscle from prenatal to postnatal periods (*Wang et al., 2015*). To identify and select better reference genes, it is necessary to evaluate the expression of more candidate genes during skeletal muscle development in both prenatal and postnatal periods. We hope to provide valuable information for gene expression analysis during different stages of skeletal muscle development in mammals, which may provide a valuable guide for the analysis of human diseases and a better understanding of muscle development.

In this study, we used transcriptome data (Supplemental Information 2) from prenatal and postnatal skeletal muscle combined with previous reports to select 15 candidate

reference genes for further analysis, including *ACTB*, *API5*, *B2M*, *GAPDH*, *RNF7*, *H3F3*, *PPIA*, *AP1S1*, *DRAP1*, *RHOA*, *RPS18*, *RPL32*, *TBP*, *WSB*, and *VAPB* (*Martino et al., 2011*; *Uddin et al., 2011*; *Wang et al., 2015*; *Zhou, Liu & Zhuang, 2014*). We collected samples of *longissimus dorsi* (LD) muscles at 26 developmental stages (including 15 prenatal and 11 postnatal periods) in Landrace pigs (a typical lean-type western breed). The expression stability of these reference genes in the porcine muscle samples was evaluated using qPCR analysis and the expression analysis programs NormFinder (*Andersen, Jensen & Orntoft, 2004*), BestKeeper (*Pfaffl et al., 2004*), and geNorm (*Vandesompele et al., 2002*).

## MATERIALS AND METHODS

### Sample collection, RNA extraction and next generation sequencing

All animals were sacrificed at a commercial slaughterhouse according to protocols approved by the Institutional Animal Care and Use Committee at the Institute of Animal Science, Chinese Academy of Agricultural Sciences (Approval number: PJ2011-012-03). *Longissimus dorsi* (LD) muscle samples were collected from Landrace fetuses on the following days post-coitum (dpc): 33, 40, 45, 50, 55, 60, 65, 70, 75, 80, 85, 90, 95, 100, and 105 dpc. LD muscle samples were collected from piglets on the following days after birth (dab): 0, 10, 20, 30, 40, 60, 80, 100, 140, 160, and 180 dab. Three biological replicates were collected at each time point, and totally 78 samples were collected. All samples were immediately frozen in liquid nitrogen and stored at –80 °C until further processing. Total RNA was extracted using TRIzol Reagent (Invitrogen, Carlsbad, CA, USA) according to the manufacturer's instructions. RNA quantity and quality was determined by the Evolution 60 UV-Visible Spectrophotometer (Thermo Scientific). RNA preparations with an $A_{260}/A_{280}$ ratio of 1.8–2.1 and an $A_{260}/A_{230}$ ratio > 2.0 were selected for this assay. RNA integrity was determined by analyzing the 28S/18S ribosomal RNA ratio on 1.5% agar gels. Only RNA preparations that resolved with three clear bands on these gels were used for the transcriptome sequencing and qPCR analysis.

### Selection of candidate reference genes

For the purpose of identifying potential reference genes during skeletal muscle development, candidate reference genes were selected according to previous studies (*Martino et al., 2011*; *Uddin et al., 2011*; *Wang et al., 2015*; *Zhou, Liu & Zhuang, 2014*). The top 15 stably reference genes in the transcriptome data of skeletal muscle at different developmental stages based on the coefficient of variation (CV) were chosen for further gene-stability evaluation by qPCR method. Lower CV values represent genes with more stable expression in our transcriptome data.

### cDNA synthesis

The cDNA synthesis was performed using the RevertAid First Strand cDNA Synthesis Kit (Thermo Scientific) according to the manufacturer's instructions for reverse transcription (RT) PCR. A mixture of 2 μg of total RNA and 1 μL of random primer was incubated at 65 °C for 5 min to dissociate the RNA secondary structure. Next, the following reaction was carried out in a total volume of 20 μL: 12 μL of the first reaction mixture,
**Table 1  Primers for the 15 candidate reference genes of RT-qPCR data analysis.**

| Gene symbol | Gene name | Amplicon length(bp) | References |
| --- | --- | --- | --- |
| API5 | Apoptosis inhibitor 5 | 82 | *Tramontana et al. (2008)* |
| AP1S1 | AP-1 complex subunit sigma-1A | 100 | *Tramontana et al. (2008)* |
| B2M | Beta-2-microglobulin | 188 | *Wang et al. (2015)* |
| DRAP1 | Down-regulator of transcription 1-associated protein1 | 157 | *Wang et al. (2015)* |
| GAPDH | Glyceraldehyde 3-phosphate dehydrogenase | 130 | *Park et al. (2015)* |
| H3F3A | H3 histone, family 3A | 181 | *Wang et al. (2015)* |
| PPIA | Peptidyl-prolylisomerase A (cyclophilin A) | 171 | *Uddin et al. (2011)* |
| RHOA | Ras homolog A | 167 | *Wang et al. (2015)* |
| RNF7 | Ring finger protein 7 | 141 | *Wang et al. (2015)* |
| RPL32 | Ribosomal protein L32 | 145 | *Wang et al. (2015)* |
| RPS18 | Ribosomal protein S18 | 74 | *Park et al. (2015)* |
| TBP | TATA box binding protein | 124 | *Martino et al. (2011)* |
| WSB | WD repeat and SOCS box-containing | 157 | *Wang et al. (2015)* |
| VAPB | VAMP-associated protein B | 100 | *Tramontana et al. (2008)* |
| ACTB | Beta-actin | 120 | *Tramontana et al. (2008)* |

4 µL of 5× RT buffer, 2 µL of 10 nM dNTP, 1 µL of RevertAid Reverse Transcriptase (200 U/µl) inhibitor, and 1 µL RiboLockRNase Inhibitor (20 U/µl). The reverse transcription reaction was performed at 25 °C for 5 min, followed by 42 °C for 1 h and 5 min at 70 °C. The cDNA was then diluted 7-fold, and stored at −20 °C until use.

## qPCR with SYBR green

Each qPCR reaction was performed in a final reaction volume of 20 µL containing 10 µL of SYBR Green Select Master Mix, 7 µL of sterile water, 0.5 µl of gene-specific primers, and 2 µL of template cDNA. The PCR amplifications were performed on a 7500 Real-Time PCR System (Applied Biosystems, Foster City, CA, USA) under the following cycling conditions: 95 °C for 5 min, followed by 40 cycles at 95 °C for 15 s and 60 °C for 45 s. Three independent individuals at each time point were used for temporal and spatial analyses. Each qPCR reaction was performed in triplicate for technical repeats. The mean quantification cycle (Cq) value was used for further analysis. The primer sequences were according to or based on those of previous reports as follows (Table 1): *B2M, RHOA, RPL32, DRAP1 RNF7, WSB,* and *H3F3* (*Wang et al., 2015*); *ACTB, AP1S1, API5,* and *VAPB* (*Tramontana et al., 2008*); *GAPDH* and *RPS18* (*Park et al., 2015*); and *TBP* and *PPIA* (*Martino et al., 2011; Uddin et al., 2011*).

## Analysis of gene expression stability

Gene expression data were transformed to relative quantities using geNorm and NormFinder. GeNorm provided a measure of gene expression stability (M) (*Vandesompele et al., 2002; McCulloch et al., 2012*)

$$M_j = \sum_{k=1}^{n} V_{jk}/n - 1$$

where:

    $Mj$ = gene stability measure,

    $Vjk$ = pairwise variation of gene $j$ relative to gene $k$,

    $n$ = total number of number of examined genes.

Lower M values represent genes with more stable expression across specimens being compared and generated a ranking of the putative reference gene expression levels from the most stable (lowest M-values) to the least stable (highest M-values). GeNorm also generated a pairwise stability measure, which can be used to evaluate the suitable number of reference genes for normalization.

NormFinder provided a stability measure (SV), identified the most stable gene, and calculated the best combination of two reference genes. This program focuses on finding the two genes with the least intra- and inter-group expression variation or the most stable reference gene in intra-group expression variation. Genes with lower stability values show a stably expressed pattern, while the higher stability values share the least stably expressed pattern.

The BestKeeper program was used to compute the geometric mean of each candidate gene's Cq value, to determine the most stably expressed genes based on correlation coefficient (r) analysis for all pairs of candidate reference genes (≤10 genes), and to calculate the percentage coefficient of variation (CV) and standard deviation (SD) using each candidate gene's crossing point (CP) value (the quantification cycle value; Cq). In the BestKeeper program, genes with higher r values and lower CV and SD values are more stable reference genes.

## RESULTS

### Expression analysis of candidate reference genes in developing skeletal muscle

We performed qPCR assays to measure the expression levels of 15 candidate reference genes (*ACTB*, *API5*, *B2M*, *GAPDH*, *RNF7*, *H3F3*, *PPIA*, *AP1S1*, *PPIA*, *RHOA*, *RPS18*, *RPL32*, *TBP*, *WSB*, and *VAPB*) in the LD muscle samples at 15 embryonic stages (33, 40, 45, 50, 55, 60, 65, 70, 75, 80, 85, 90, 95, 100, and 105 dpc) and 11 postnatal stages (0, 10, 20, 30, 40, 60, 80, 100, 140, 160, and 180 dab) of Landrace pigs. To minimize experimental error, triplicate amplifications were performed for individual experiments. The Cq values were computed to quantify the candidate reference gene expression levels. A higher Cq value means lower gene expression levels. Analysis of gene expression stability was based on the Cq values generated by qPCR (Fig. 1). Among all the tested genes, *GAPDH* had the lowest mean Cq value (15.94) and *AP1S1* had the highest mean Cq value (26.36). All candidate reference genes were abundantly expressed in skeletal muscle and showed wide variations in expression levels at different developmental stages. Therefore, it was necessary to evaluate gene expression stability and determine the suitable number of reference genes for accurate gene expression profiling in developing skeletal muscle.
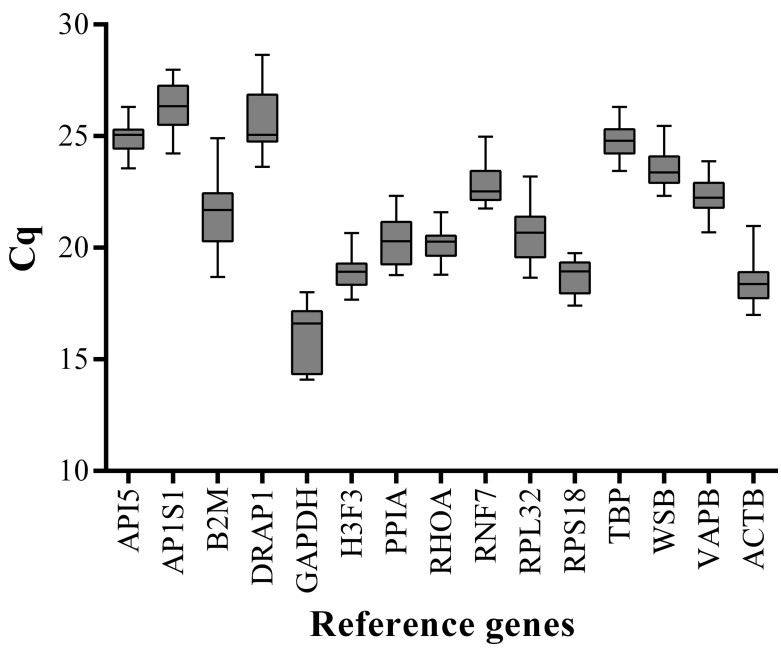

**Figure 1** **Box-and-whisker plot displaying the range of Cq values for each reference gene.** The median is marked by the middle line in the box.

## GeNorm analysis of candidate reference gene expression stability

We calculated the gene expression stability values (M value) for the 15 candidate reference genes using the geNorm program. Genes with lower M values have more consistent expression levels. The M values of the candidate genes are presented in Fig. 1. When all developmental stages were analyzed as one data set. The results revealed that *API5* and *H3F3* had the lowest M values, whereas *GAPDH* had the highest M value. This indicated that *API5* and *H3F3* were the most stably expressed gene pair of the 15 candidate reference genes, whereas *GAPDH* had the most variable expression (Fig. 2) in developing skeletal muscle across prenatal and postnatal periods. In the prenatal muscle samples, *API5* and *RPS18* expression was the most stable, whereas *GAPDH* and *DRAP1* expression was the most variable (Fig. 3). When only postnatal muscle samples were analyzed, *API5* and *RPS18* expression was the most stable, whereas *B2M* expression was the most variable (Fig. 4). The geNorm analysis demonstrated that *GAPDH* was the most variably expressed gene in all developmental periods, suggesting that *GAPDH* was not a suitable reference gene for gene expression analysis in developing skeletal muscle. By contrast, the stability of *API5*, *RPS18*, and *H3F3* expression suggested that they were suitable reference genes to use as internal controls. When gene expression was analyzed in developing skeletal muscle across all tested prenatal and postnatal periods, *API5* expression was the most suitable to use as a reference gene for normalization analysis in expression profiling studies.

One single reference gene might not provide sufficient control for gene expression analyses in developing skeletal muscle. Therefore, we used geNorm to analyze the optimal number of reference genes required to obtain reliable results from RT-qPCR studies.

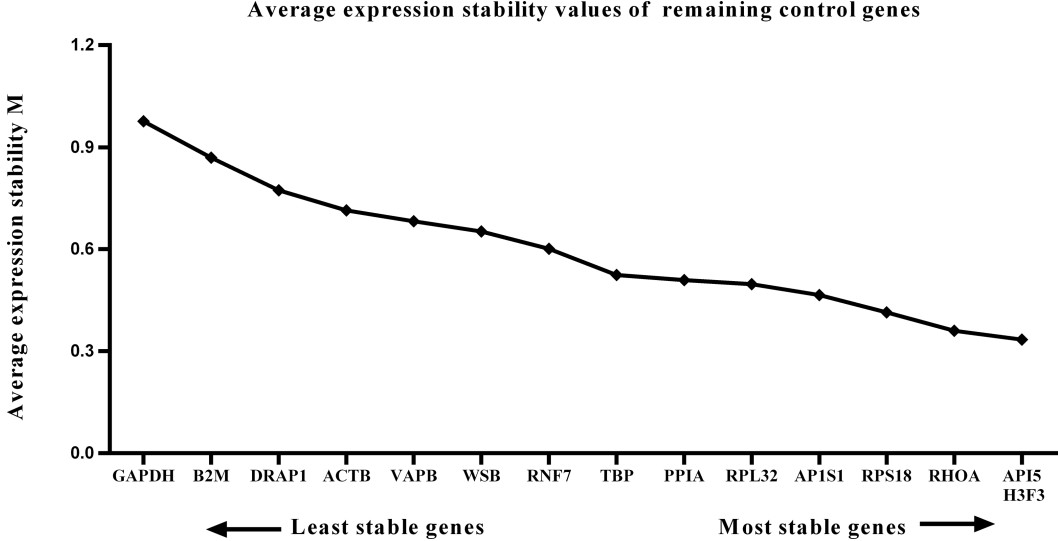

**Figure 2  Average expression stability (M) of 15 candidate reference genes and the best combination of two genes were calculated for 26 developmental periods.** Lower M values indicate more stable expression.

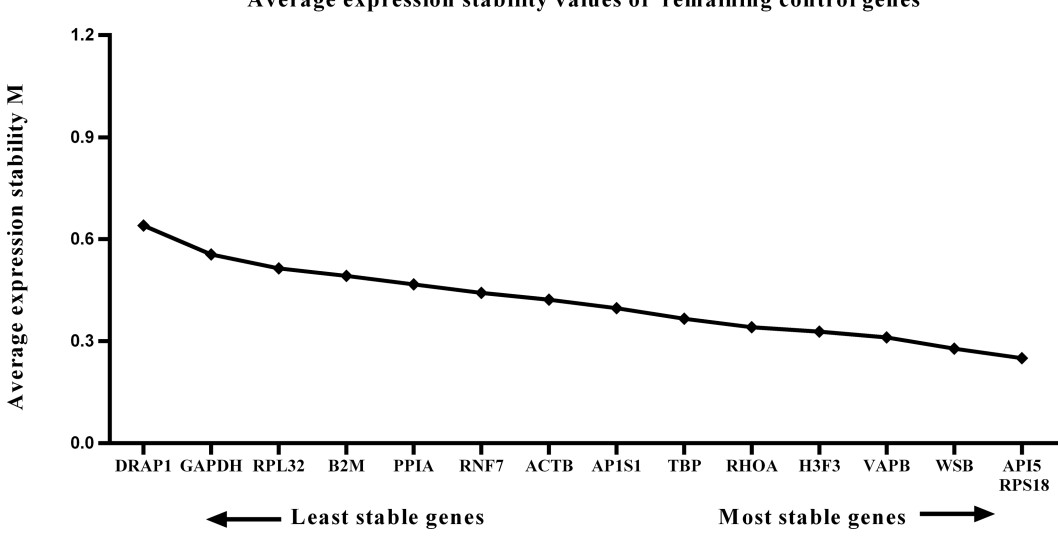

**Figure 3  Average expression stability (M) of 15 candidate reference genes and the best combination of two genes were calculated for the prenatal periods.** Lower M values indicate more stable expression.

GeNorm was used to calculate the average pairwise variation (V) value between two sequential normalization factors; it has a cut-off value of 0.15 for the pairwise variation according to the previous study (*Wang et al., 2015*), below which the inclusion of an additional reference gene is not required for reliable normalization of qPCR analyses. When all developmental stages were analyzed together, the $V_n/V_{n+1}$ value ranged from 0.059 to 0.111, which were all lower than the cut-off value of 0.15 (Fig. 5). These results indicated that two reference genes were optimal for gene expression analysis of all tested developmental

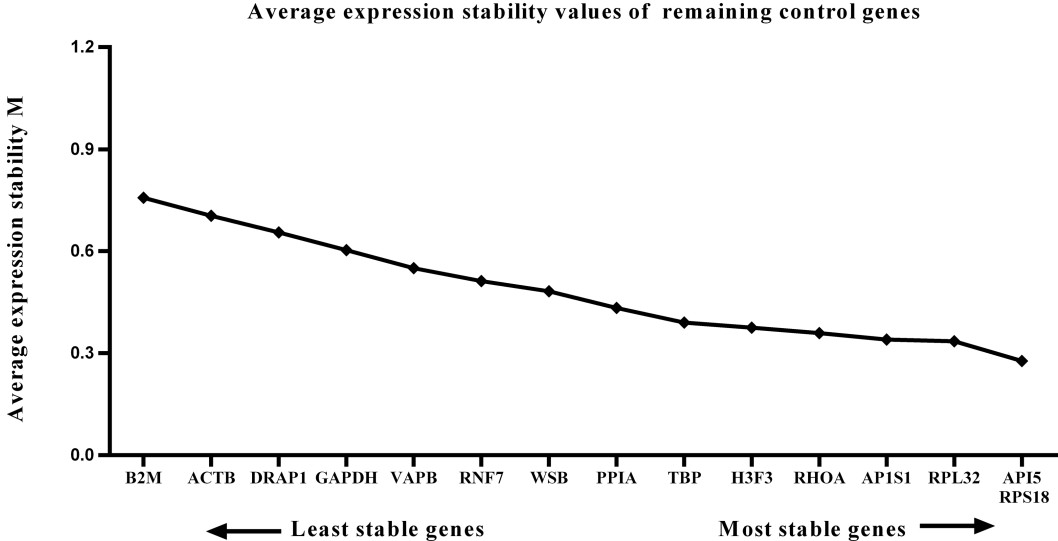

**Figure 4** **Average expression stability (M) of 15 candidate reference genes and the best combination of two genes were calculated for the postnatal period.** Lower M values indicate more stable expression.

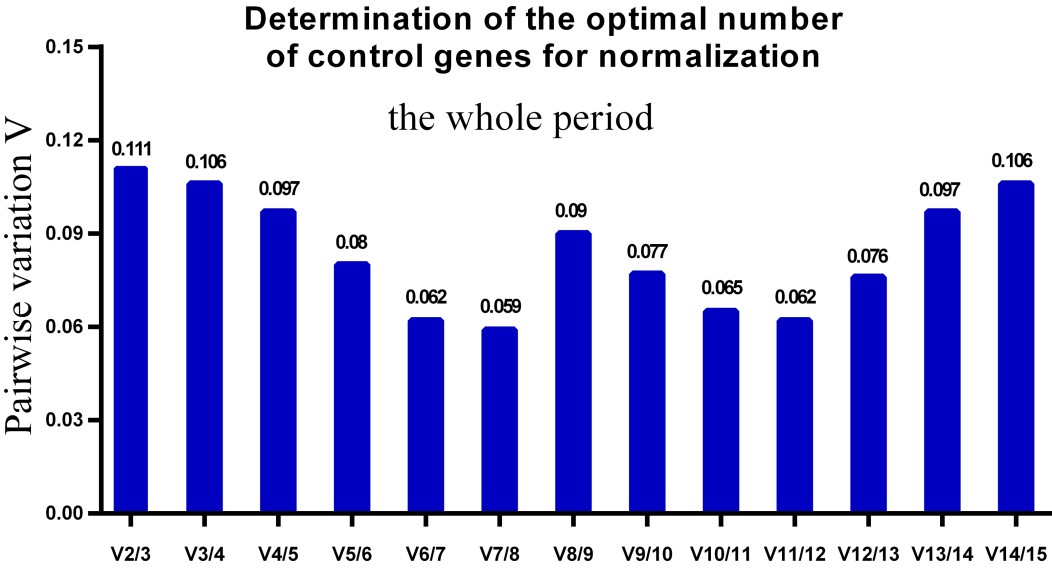

**Figure 5** **Determination of the optimal number of reference genes for normalization in the whole tested period.** GeNorm was used to calculate the normalization factor (NF) from at least two genes; the variable V defines the pair-wise variation between two sequential NF values.

periods. The results were similar for gene expression analysis of the embryonic data set (Fig. 6), and two reference genes were sufficient for analysis. By contrast, the V value decreased significantly with the addition of reference genes in the postnatal data set, although all values were lower than 0.15 (Fig. 7). These results indicated that the three most stable reference genes were required for accurate normalization of gene expression data for the postnatal period.

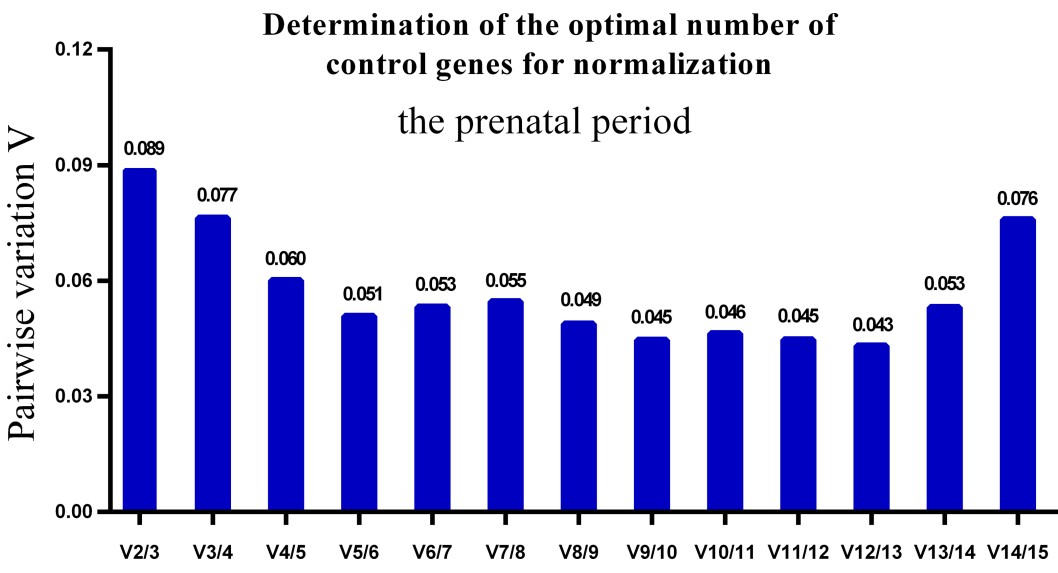

**Figure 6 Determination of the optimal number of reference genes for normalization in prenatal periods.** GeNorm was used to calculate the normalization factor (NF) from at least two genes; the variable V defines the pair-wise variation between two sequential NF values.

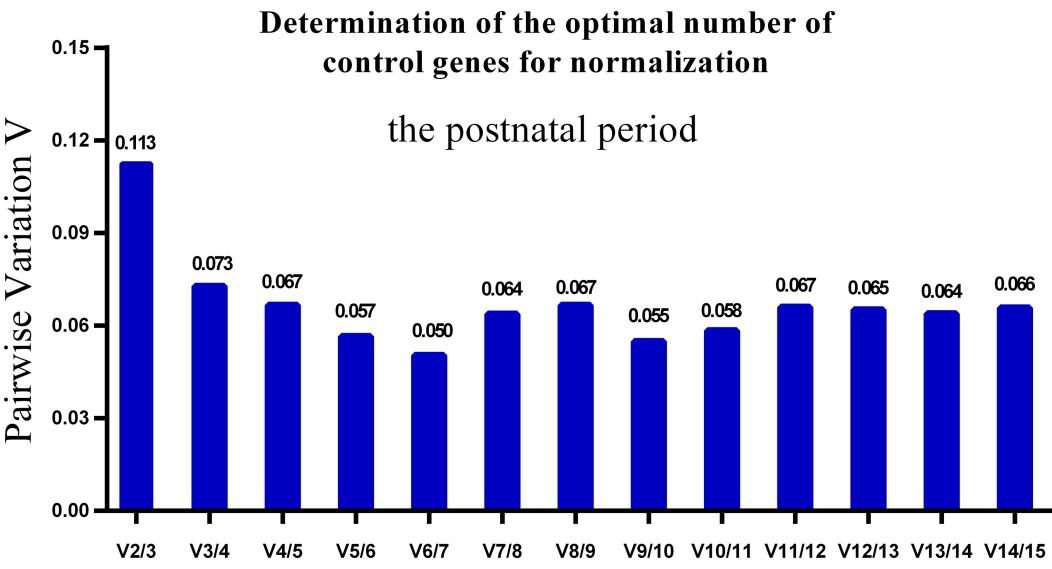

**Figure 7 Determination of the optimal number of reference genes for normalization in postnatal periods.** GeNorm was used to calculate the normalization factor (NF) from at least two genes; the variable V defines the pair-wise variation between two sequential NF values.

## NormFinder analysis of candidate reference gene expression stability

Next, we used NormFinder to rank the most stable and the least stable genes by calculating the gene expression stability value and standard error. NormFinder analyses showed that *API5* was the most stable reference gene with the lowest stability value (SV = 0.088) in all tested developmental periods (Table 2). *API5* was the most stable reference gene in

**Table 2** Calculations of gene stability valueby NormFinder program.

| Gene stability value calculations by NormFinder. | | | | | |
| --- | --- | --- | --- | --- | --- |
| The whole period | | Prenatal period | | Postnatal period | |
| Gene name | Stability value | Gene name | Stability value | Gene name | Stability value |
| AIP5 | 0.088 | RPS18 | 0.101 | AIP5 | 0.167 |
| H3F3 | 0.150 | AIP5 | 0.112 | RHOA | 0.173 |
| RHOA | 0.242 | H3F3 | 0.146 | H3F3 | 0.193 |
| RNF7 | 0.354 | WSB | 0.172 | RPL32 | 0.236 |
| PPIA | 0.373 | RHOA | 0.176 | AP1S1 | 0.238 |
| WSB | 0.378 | VABP | 0.233 | TBP | 0.284 |
| RPS18 | 0.394 | AP1S1 | 0.25 | RNF7 | 0.303 |
| VABP | 0.422 | RNF7 | 0.281 | RPS18 | 0.308 |
| RPL32 | 0.429 | PPIA | 0.293 | WSB | 0.351 |
| TBP | 0.434 | TBP | 0.31 | PPIA | 0.368 |
| AP1S1 | 0.468 | B2M | 0.339 | VABP | 0.501 |
| ACTB | 0.546 | ACTB | 0.352 | GAPDH | 0.501 |
| DRAP1 | 0.700 | RPL32 | 0.369 | DRAP1 | 0.541 |
| B2M | 0.865 | GAPDH | 0.549 | ACTB | 0.598 |
| GAPDH | 1.096 | DRAP1 | 0.786 | B2M | 0.677 |

postnatal periods, whereas *RPS18* was the most stable reference gene in prenatal periods (Table 2).

**BestKeeper analysis of candidate reference gene expression stability**

Then, BestKeeper program was used to evaluate the reference gene expression stability. We used BestKeeper to identify the optimal reference genes on the basis of the correlation coefficient ($r$), CV, and SD values (Table 3). The program can calculate $r$ values for up to 10 genes. Therefore, we selected the top 10 candidate genes based on the previous results. In the BestKeeper program, genes with higher $r$ values ($\geq 0.900$) and lower CV and SD values are considered as stable and suitable reference genes. In all tested developmental periods, *API5* expression had the lowest CV value (2.09) and almost the lowest SD value (it was slightly larger than that of *RPL32* expression). Therefore, we propose that *API5* is the most suitable reference gene for expression analysis of developing skeletal muscle during the tested prenatal and postnatal stages. *API5* also was selected as the most stable gene during the postnatal period, whereas *VAPB* was the most stable gene for the analysis of developing skeletal muscle during the embryonic period.

## DISCUSSION

Studies of muscle development are important to improve meat production, to understand human diseases like diabetes (*Li et al., 2016a*; *Nixon et al., 2016*; *Obata et al., 2016*; *Zabielski et al., 2016*) and obesity (*Costa Junior et al., 2016*; *Fonvig et al., 2015*; *Putti et al., 2015*; *Thivel et al., 2016*) due to the important role of skeletal muscle in lipid and energy metabolism. Many studies investigated the mechanism of skeletal muscle development by

**Table 3  Expression stability analysis of the reference genes by BestKeeper.**

| whole period | API5 | AP1S1 | H3F3 | RHOA | RPL32 | PPIA | RNF7 | RPS18 | TBP | WSB |
|---|---|---|---|---|---|---|---|---|---|---|
| $n$ | 26 | 26 | 26 | 26 | 26 | 26 | 26 | 26 | 26 | 26 |
| geo Mean | 24.92 | 26.36 | 20.55 | 18.91 | 20.13 | 20.28 | 22.81 | 18.70 | 24.79 | 23.57 |
| std dev | 0.52 | 0.81 | 0.90 | 0.55 | 0.51 | 0.92 | 0.72 | 0.69 | 0.59 | 0.70 |
| CV | 2.09 | 3.05 | 4.38 | 2.91 | 2.52 | 4.55 | 3.15 | 3.69 | 2.39 | 2.96 |
| $r$ | 0.911 | 0.924 | 0.982 | 0.935 | 0.950 | 0.937 | 0.603 | 0.912 | 0.793 | 0.524 |
| $p$-value | 0.001 | 0.001 | 0.001 | 0.001 | 0.001 | 0.001 | 0.001 | 0.001 | 0.001 | 0.006 |
| **prenatal period** | API5 | AP1S1 | H3F3 | RHOA | PPIA | RNF7 | RPS18 | TBP | WSB | VAPB |
| $n$ | 15 | 15 | 15 | 15 | 15 | 15 | 15 | 15 | 15 | 15 |
| geo Mean | 24.93 | 25.94 | 22.48 | 25.54 | 23.12 | 20.24 | 18.36 | 24.47 | 23.95 | 22.68 |
| std dev | 0.64 | 0.72 | 0.84 | 1.40 | 0.83 | 1.04 | 0.63 | 0.49 | 0.71 | 0.55 |
| CV | 2.57 | 2.79 | 3.75 | 5.48 | 3.57 | 5.11 | 3.41 | 2.00 | 2.94 | 2.42 |
| $r$ | 0.97 | 0.95 | 0.892 | 0.835 | 0.921 | 0.99 | 0.971 | 0.838 | 0.930 | 0.904 |
| $p$-value | 0.001 | 0.001 | 0.001 | 0.001 | 0.001 | 0.001 | 0.001 | 0.001 | 0.001 | 0.001 |
| **postnatal period** | API5 | AP1S1 | H3F3 | RHOA | RPL32 | PPIA | RNF7 | RPS18 | TBP | WSB |
| $n$ | 11 | 11 | 11 | 11 | 11 | 11 | 11 | 11 | 11 | 11 |
| geo Mean | 24.90 | 26.94 | 20.98 | 20.38 | 18.98 | 20.55 | 22.39 | 19.17 | 25.24 | 23.05 |
| std dev | 0.36 | 0.52 | 0.54 | 0.30 | 0.29 | 0.75 | 0.29 | 0.36 | 0.51 | 0.38 |
| CV | 1.43 | 1.93 | 2.55 | 1.49 | 1.53 | 3.66 | 1.30 | 1.90 | 2.01 | 1.63 |
| $r$ | 0.905 | 0.962 | 0.963 | 0.856 | 0.846 | 0.912 | 0.277 | 0.894 | 0.872 | 0.327 |
| $p$-value | 0.001 | 0.001 | 0.001 | 0.001 | 0.001 | 0.001 | 0.412 | 0.001 | 0.001 | 0.325 |

performing gene expression analysis (*Krist et al., 2015*; *Wang, Xiao & Wang, 2016*; *Zhang et al., 2016a*; *Zhang et al., 2015*). However, it is crucial to select accurate reference genes to normalize target gene expression levels during skeletal muscle development in mammals. A number of different genes have been commonly used for normalizing gene expression in skeletal muscle, including *ACTB* and *GAPDH*. It was assumed that the expression of these genes was perfectly stable. However, many experiments have shown that these reference genes have variable expression levels in developing skeletal muscle (*Wang et al., 2015*; *Selvey et al., 2001*).

Many researchers have studied the suitable reference genes in pig skeletal muscle. For example, *Feng et al. (2010)* found that *PPIA* and *HPRT* were the most stable reference genes for gene expression studies in LD muscles of postnatal Yorkshire pigs. *Wang et al. (2015)* reported that *DRAP1* and *RNF7* were the most appropriate combination of reference genes to normalize gene expression in postnatal developing muscle of Yorkshire pigs. These previous studies tested only a limited number of candidate reference genes and a limited number of developmental stages. By contrast, we selected many different candidate reference genes and tested gene expression in many developmental stages (essentially covering the whole period of the pig lifespan under investigation). Therefore, the results of our study are more robust and accurate.

We selected 15 candidate reference genes and performed qPCR analysis of their mRNA expression. The results analyzed by three different algorithms (NormFinder, BestKeeper, and geNorm) showed that apoptosis inhibitor 5 (*API5*) was the best candidate reference gene for normalizing target gene expression in developing skeletal muscle across the tested prenatal and postnatal periods. *API5* is highly conserved across species from microorganisms to plants and animals (*Li et al., 2011*; *Mayank et al., 2015*; *Noh et al., 2014*). *AP15* has an important role in negative regulation of apoptotic processes in fibroblasts (*Kim et al., 2000*; *Noh et al., 2014*). This gene encodes an inhibitory protein that prevents apoptosis after growth factor deprivation. The *API5* protein suppresses apoptosis induced by the transcription factor E2F1, and interacts with and negatively regulates Acinus, a nuclear factor involved in apoptotic DNA fragmentation. The *API5* gene is involved in many human diseases including diabetes and cancers (*Cho et al., 2014*; *Noh et al., 2014*; *Peng et al., 2015*; *Ramdas et al., 2011*). However, *API5* has not been reported to be involved in skeletal muscle development. We hypothesize that *API5* may play an important role in skeletal muscle development as a housekeeping gene, based on the observed constitutive expression across prenatal and postnatal developing skeletal muscle in pigs.

*Park et al. (2015)* examined the expression stability of different genes in various tissues, and found that *PPIA*, *TBP*, *RPL4*, and *RPS18* were the suitable reference genes in Landrace pigs (*Park et al., 2015*). Our results are consistent with these conclusions. The combination of *DRAP1* and *WSB2* is appropriate for the whole tested developmental period in Tongcheng pigs (an obese-type Chinese native breed) (*Wang et al., 2015*), whereas our study showed that *DRAP1* and *RNF7* were unsuitable as reference genes in prenatal and postnatal developmental periods in Landrace pigs. These differences may be caused by the developmental stages tested, or that we tested more developmental stages in our study. *H3F3* was reported as a suitable reference gene in the prenatal period in Tongcheng pigs, which was consistent with the results of our study.

We previously reported that *RPL32*, *RPS18*, and *H3F3* were the most stable reference genes in 33, 65, and 90 dpc skeletal muscle in Landrace pigs (*Zhang et al., 2012*). The current results also identify these genes as suitable reference genes for normalizing target gene expression in developing skeletal muscle during the prenatal periods. We selected candidate reference genes during skeletal muscle development based on transcriptome data and previous studies, which might provide a new clue for evaluating the stability of candidate reference genes. Combined with multiple methods, our evaluated results would be more precious and accurate. However, our present work only evaluated these candidate references on the Landrace pigs. Further studies are needed to further evaluate the stability of these genes during skeletal muscle development at other pig breeds and mammals.

## CONCLUSION

Our study evaluated the expression stability of 15 candidate reference genes in LD skeletal muscle across 26 prenatal and postnatal developmental periods in Landrace pigs. We found that the commonly used reference genes (*GAPDH* and *ACTB*) were not suitable as reference genes for skeletal muscle development. Our results showed that *API5*, a newly

discovered reference gene, was the most suitable reference gene for all tested periods and muscle samples. *RPL32*, *RPS18*, *VAPB*, and *H3F3* also were suitable as reference genes in developing skeletal muscle. Our data provide a guide for choosing appropriate reference genes for studies on skeletal muscle development and diseases in humans and other mammals.

### Funding

This work was supported by the National Key Project (2016ZX08009003-006-003), the National Natural Science Foundation of China (31372295 and 31330074), and the Agricultural Science and Technology Innovation Program (ASTIP-IAS16). The funders had no role in study design, data collection and analysis, decision to publish, or preparation of the manuscript.

### Grant Disclosures

The following grant information was disclosed by the authors:
National Key Project: 2016ZX08009003-006-003.
National Natural Science Foundation of China: 31372295, 31330074.
Agricultural Science and Technology Innovation Program: ASTIP-IAS16.

### Competing Interests

The authors declare there are no competing interests.

### Author Contributions

- Guanglin Niu conceived and designed the experiments, performed the experiments, analyzed the data, contributed reagents/materials/analysis tools, wrote the paper, prepared figures and/or tables, reviewed drafts of the paper.
- Yalan Yang analyzed the data, contributed reagents/materials/analysis tools, wrote the paper, reviewed drafts of the paper.
- YuanYuan Zhang performed the experiments, contributed reagents/materials/analysis tools, reviewed drafts of the paper.
- Chaoju Hua, Zishuai Wang and Kui Li reviewed drafts of the paper.
- Zhonglin Tang conceived and designed the experiments, wrote the paper, reviewed drafts of the paper.

### Animal Ethics

The following information was supplied relating to ethical approvals (i.e., approving body and any reference numbers):

Animal euthanasia was performed according to protocols approved by Institutional Animal Care and Use Committee at the Institute of Animal Science, Chinese Academy of Agricultural Sciences Approval number: PJ2011-012-03.

### Data Availability

The raw data has been supplied as Supplemental Files.

## Supplemental Information

Supplemental information for this article can be found online at http://dx.doi.org/10.7717/peerj.2428#supplemental-information.

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
