# Peer review of "Identifying suitable reference genes for gene expression analysis in developing skeletal muscle in pigs"

_PeerJ, doi:10.7717/peerj.2428_

## Round 0.1 · original submission · Major Revisions

Experts in the field have carefully reviewed your manuscript titled: "Identifying suitable reference genes for gene expression analysis in developing skeletal muscle in pigs". While your paper is of interest, major concerns were raised that require your attention before the paper can be accepted in its final form. Accordingly, I invite you to respond to the reviewers’ comments and recommendations. A decision on acceptability of your manuscript will be made only after the
revised version has been reevaluated.

Reviewer 1 ·

Basic reporting

The choice of reference genes is a critical issue today. The authors put efforts into the extend the reference gene candidates. It is publishable if they could answer the following concerns.
1. I don't question their experimental skills, the qOCR is a routine method in most of biological lab. I really concern whether they understand well the evaluation method. The main method they used in the manuscript is GeNorm software and BestKeeper program, they don't any explanation of the methods, how to figure out the gene expression stability (M). They should give main idea and formulas.
2. The figure 5, 6 and 7 are not labeled Y axis clearly.
3. The English should be edited.

Experimental design

They only did triplicates for each genes. It is not clear, how many samples. The plot is the boxplot based on how many samples, if only 3 times of qPCR, it doean't make sense. Therefore, the sample size is very important. They should make very clear about sample size, biological replicates and technical repeats.

Validity of the findings

The study needs further work to claim the API5 is a valuable for another reference gene.

Reviewer 2 ·

Basic reporting

The manuscript requires language polishing. Relevant previous studies have not been examined and cited.
The objectives of the study should be explicitly written.

Experimental design

The last paragraph of introduction section describes about use of transcriptome data for analysis, which is not indicated in the methods section.
Insufficient description of materials and methods such as:
a. No details of geNorm and NormFinder, bestkeeper programmes used by authors are described including its source.
b. no details on number of animals, replicates used for the study.

Validity of the findings

a. The reason for choosing cut off of 0.15 for variance ratio was chosen by authors, a sentence on this aspect may be included.
b. The authors state that "We previously reportedthat RPL32, RPS18, and H3F3were the most stable reference genes in 33, 65, and 90 dpc skeletal muscle in Landrace pigs(Zhang et al. 2012).The current results also identify these genes as suitable reference genes for normalizing target gene expression in developing skeletal muscle during the prenatal period".
c. Comments on limitations of the programmes/experimental design used by authors should be indicated in the manuscript.

The authors should be able to demonstrate how the present studies improved understanding on selection of reference genes.

Additional comments

The authors have examined gene expression analysis in developing skeletal muscle in pigs of identification of suitable reference genes . The study has little novelty. Earlier studies (BMC Mol Biol. 2007; 8: 67, Journal of Animal Science and Biotechnology20123:36) that has specifically examined stability of gene expressions should be cited.

---

## Round 0.2 · accepted · Accept

Your revised manuscript has been reevaluated. The revised manuscript has answered all reviewers' concerns. It now meets the requirements for publication.